# Promising Lead Compounds in the Development of Potential Clinical Drug Candidate for Drug-Resistant Tuberculosis

**DOI:** 10.3390/molecules25235685

**Published:** 2020-12-02

**Authors:** Saad Alghamdi, Shaheed Ur Rehman, Nashwa Talaat Shesha, Hani Faidah, Muhammad Khurram, Sabi Ur Rehman

**Affiliations:** 1Laboratory Medicine Department, Faculty of Applied Medical Sciences, Umm Al-Qura University, Mecca 24321, Saudi Arabia; ssalghamdi@uqu.edu.sa; 2Department of Pharmacy, Abasyn University Peshawar, Khyber Pakhtunkhwa 25000, Pakistan; sabi.rehman@abasyn.edu.pk; 3Regional Laboratory, Directorate of Health Affairs Makkah, Mecca 24321, Saudi Arabia; nshesha@moh.gov.sa; 4Microbiology Department, Faculty of Medicine, Umm Al-Qura University, Mecca 24321, Saudi Arabia; hsfaidah@uqu.edu.sa

**Keywords:** drug-resistant tuberculosis, isoniazid derivatives, coumarin derivatives, antimicrobial peptides

## Abstract

According to WHO report, globally about 10 million active tuberculosis cases, resulting in about 1.6 million deaths, further aggravated by drug-resistant tuberculosis and/or comorbidities with HIV and diabetes are present. Incomplete therapeutic regimen, meager dosing, and the capability of the latent and/or active state tubercular bacilli to abide and do survive against contemporary first-line and second line antitubercular drugs escalate the prevalence of drug-resistant tuberculosis. As a better understanding of tuberculosis, microanatomy has discovered an extended range of new promising antitubercular targets and diagnostic biomarkers. However, there are still no new approved antitubercular drugs of routine therapy for several decades, except for bedaquiline, delamanid, and pretomanid approved tentatively. Despite this, innovative methods are also urgently needed to find potential new antitubercular drug candidates, which potentially decimate both latent state and active state *mycobacterium tuberculosis*. To explore and identify the most potential antitubercular drug candidate among various reported compounds, we focused to highlight the promising lead derivatives of isoniazid, coumarin, griselimycin, and the antimicrobial peptides. The aim of the present review is to fascinate significant lead compounds in the development of potential clinical drug candidates that might be more precise and effective against drug-resistant tuberculosis, the world research looking for a long time.

## 1. Introduction

Mycobacterium is a genus of Actinobacteria, acid-fast, aerobic, and nonmotile bacteria. This genus contains opportunistic pathogens known to cause serious diseases, especially in the clinical setting. *Mycobacterium tuberculosis*, the etiological agent of tuberculosis is one of the leading causes of morbidity and mortality in humans [1]. In 80–90% of the cases, these aerobic bacilli enter the lungs, while other extrapulmonary organs, primarily lymph nodes, bones, joints, and the genitourinary system are affected as well [2]. The primary route of transmission of the bacilli into the human lungs is via aerosol droplets [3,4,5].

According to WHO report, globally, about 10 million active tuberculosis cases, resulting in about 1.6 million deaths, further aggravated by drug-resistant tuberculosis and/or comorbidities with diabetes and HIV are present [6]. Drug-resistant tuberculosis is a growing threat to public health. Around half a million new rifampicin-resistant tuberculosis cases (78%—multidrug-resistant tuberculosis) emerged in 2018. Globally, 3.4% of new cases of tuberculosis had multidrug-resistant tuberculosis or rifampicin-resistant tuberculosis (MDR/RR-TB) and 18% of previously treated patients [6,7]. In 2018, an estimated 1.2 million deaths from tuberculosis were reported in HIV negative persons, while further 251,000 deaths among HIV-positive cases [6] were observed. Tuberculosis coinfection with HIV also raises the risk of relapse per year by up to 10%. Therefore, in patients with HIV coinfection, tuberculosis is the primary cause of death [7].

Antibiotic therapy prolongation contributes to the development of antibiotic resistance. Drug-resistant tuberculosis is an emerging public health threat in many countries of the world. Moreover, the extended duration of antibiotic treatment raises the risk of noncompliance, adverse effects, and drug toxicity. Therefore, it is urgent to find a way to strengthen the current antibiotic therapy regimen against drug-resistant tuberculosis [8]. The process of novel drug development to fight out the resistant strains of tubercular infections has been considered for a long time, and some of the significant candidates are in various stages of clinical trials [9]. Unfortunately, the drug development and approval processes are too lengthy, and these agents are often generally not as effective as expected. In tuberculosis chemotherapy, the other major problem is the relative refractory of mycobacteria to killing by antibiotics or immune systems and the development of resistant strains [10]. Recent advancements in molecular biology and mycobacterial disease immunopathogenesis have contributed to substantial studies in such research areas. Numerous vaccines are manufacturing for prevention or therapeutic purposes [11,12]. We should, however, wait in the coming years for accessible and reliable vaccines. The problematic scenario is the relationship among the host cells and tubercle bacilli afterwards the primary infection occurs, which results in latent tuberculosis infection in most cases. As described earlier, the human body can contain the disease in >80% of the cases, and the bacilli remain dormant throughout infected peoples’ entire lives. This phenomenon has been the topic of extensive research [13,14,15]. It is worth mentioning that the WHO, international organizations, educational agencies, foundations, and sponsors are working together to resolve dissimilar facets of the topic and to accomplish the anticipated goals in the approaching periods.

The development of drug-resistant strains of *mycobacterium tuberculosis* illustrates the importance and demand for an early identification of drug-resistant strains, exploring new targets for drug sensitivity, customized treatment plans, and more effective medical interventions. Literature shows several studies, incorporating bioinformatics and proteomics approaches that clearly indicate the potential drug targets and an early diagnostic against drug-resistant strains [16,17,18,19,20]. To tackle the alarming condition of antimicrobial resistance, pathogen-centric approach covering novel chemotherapeutics and novel diagnostic pathways, along with host targeted therapeutics (i.e., host immune system modulators to treat pathogenesis), must be appraised [21,22,23,24,25]. The efficacy of novel chemotherapeutic agents (i.e., delamanid and bedaquiline), which currently have approval from USFDA, are now compromised by the successional pathogen tolerance strategies [26,27,28,29,30]. Novel antitubercular repurposed drugs as combinational treatment solutions (new anti-TB drug schedules) and host-directed therapeutics may be measured to tackle the antibiotic resistance, which is a major problem to tuberculosis management. To combat the antibiotic resistance, the key problem in tuberculosis management, we have to identify the most promising lead compounds among new emerging antitubercular agents and conclude these compounds to clinical trials as potential antitubercular drug candidates, along with considering the host-targeted therapeutics.

The use of accelerated molecular microbiological techniques will be the near future of tuberculosis management [12]. Drug tests for susceptibility tests may be either phenotypic or genotypic. Phenotypic tests for drug susceptibility are commonly used with prime focus on measuring bacterial growth with first-line, e.g., isoniazid, ethambutol (EMB), rifampicin, and streptomycin [SM]), or second-line antitubercular medicines at defined concentrations. This method has certain advantages that it is fast and economical, but has disadvantage of substantial interruptions and is not reliable for use of ethambutol and other medicine. Genotypical strategies take benefit of particular transmutations relevant to the response against specific medications [13]. Molecular approaches should be conducted using sputum or any other samples and can yield results in a very short period of time (including drug susceptibility). The most used strategies are discussed below:GenXpert MTB/RIF is an assay for nucleic acid amplification in sputum sample that analyses DNA and RIF resistance for the presence of *mycobacterium tuberculosis*. It is a very simple and reproducible procedure with 90% sensitivity and 99% accuracy. The GenXpert is an automated assay and requires no laboratory arrangements. Rifampicin resistance detection is also used for the prediction of MDR-tuberculosis with isoniazid resistance (in most of the cases) [14]. As an initial diagnostic examination, the WHO recently proposed GenXpert for patients with HIV supposed to have tuberculosis or for those who are at risk for rifampicin resistance and/or MDR tuberculosis. GenXpert (MTB/RIF) Ultra is theoretically facilitating, more precise, and sensitive bedside testing that can enhance tuberculosis detection in smear-negative patients, and similar assays are currently under development.Line probe assay (LPA), approved by WHO, is family of DNA strip-based tests for swift recognition of first- and second-line antitubercular agents drug resistance [31]. It can also be used for testing culture isolates along with direct testing of acid-fast bacilli as well as smear-positive and -negative sputum specimens. LPA can determine the frequently identified mutations in resistant strains [14,15].

## 2. Compounds with Promising Antimycobacterial Potentials

Ethambutol, isoniazid, pyrazinamide, rifampicin, amikacin, cycloserine, rifabutin, amoxicillin-clavulanic acid, streptomycin, clofazimine, bedaquiline, aminosalicylic acid, delamanid, ethionamide, moxifloxacin, levofloxacin, linezolid, meropenem, and rifapentine are currently included in the list of essential WHO antituberculosis medications (WHO 2019). The three newly approved bedaquiline, delamanid, and pretomanid showed adverse events and complications when concomitantly used with other antitubercular drugs, and so, they could not be prescribed under routine treatment. Novel antitubercular medicines are less effective, while still a very limited number of the investigational drug compounds are subjected to human clinical trials.

### 2.1. Classification of Antitubercular Drugs

In the current classification, antitubercular drugs are classified into four groups, i.e., A, B, C, and D (Table 1) [32]. In this classification, it is precisely configured to incorporate the treatment of rifampicin-resistant or MDR-tuberculosis [33].
Group-A contains fluoroquinolones (high doses of levofloxacin, moxifloxacin, and gatifloxacin). Due to their bactericidal and sterilizing efficacy and strong safety profile, these are called vital products.Group-B contains injectable products like (streptomycin, kanamycin, amikacin, and capreomycin) that are incredibly bactericidal but have a lower safety rating than drugs in Group A.Ethionamide, cycloserine/terizidone, clofazimine, prothionamide, and linezolid are in Group-C. Given increasing proof of their effectiveness and tolerability, these medications are recommended as vital second-line medicines for multidrug-resistant tuberculosis.Group-D is classified into three subgroups: D1: large-dose isoniazid, ethambutol, and pyrazinamide; D2: delamanid and bedaquiline; and D3: para-aminosalicylic acid, meropenem, cilastatin-imipenem, clarithromycin, and clavulanate-amoxicillin.

Some of the emerging novel drugs have the ability to be used as adjunct in the abovementioned antitubercular regime. The inclusion of such drugs in tuberculosis therapeutics will steer synergistic effects and can lead to improved outcomes by increasing effectiveness against resistant *mycobacterium tuberculosis* strains (like isoniazid hydrazides and nitazoxanide) or by negating mechanisms of the resistance (like dihydropyridomycin). Furthermore, the development of isoniazid hydrazides and primaquine derivatives enhance the probability of improving patient compliance by decreasing adverse drug reactions and complications.

It is promising to see that many potential new candidate compounds being proposed are in pipeline to be significantly used as either monotherapy or combination therapy options to eradicate tuberculosis. However, a variety of prospective drug candidates having direct and/or indirect antitubercular action (just like sildenafil, mefloquine, cilostazol, tizoxanide, metronidazole, entacapone, tolcapone, and ibuprofen) are reported to have high efficacy during in vitro studies as mycobacterial monotherapy or in adjunctive action [31,34]. These compounds need to be tested in vivo and, later on, in human clinical trials. In enlightenment of the above statement, it is crucial to participate in the recent antitubercular drug discovery campaign that focuses on designing drug regimens capable of shortening treatment duration as well as fight out complicated drug resistance tuberculosis. Thus, it is important to highlight the potential antitubercular drug candidates among others, so, to pursue further research and in vivo studies based on drug safety and efficacy against drug-resistant mycobacterium strains.

### 2.2. Promising Novel Chemotherapeutics

#### 2.2.1. Isoniazid Lead Derivatives

Isoniazid (isonicotinic hydrazide, pyridine-4-carbohydrazide, and INH) is one of the most significant and efficient first-line antitubercular drugs in tuberculosis therapeutics around the world. Since 1952, isoniazid has been extensively used as a prodrug, which is activated by catalase-peroxidase (KatG), the mycobacterial multifunctional enzyme. The inhibition of mycolic acids synthesis, a distinct building block in the mycobacterium cell wall and accountable for its prominent lipophilic nature, is the main target of isoniazid. The activated isoniazid molecule is responsible for suppressing biosynthesis of mycolic acid by enoyl-ACP reductase inhibition, an enzyme elaborated in the biosynthesis of fatty acid in the existence of NAD+ or NADH [35]. So, the enoyl-ACP reductase is considered one of the finest-validated targets in the therapeutics of tuberculosis. Isoniazid resistance is mostly mediated by the mutations of the katG gene, which leads to the drug activation incapability, one of the prominent factors responsible for *mycobacterium tuberculosis* resistance in the clinical settings [36]. The mutation rate liable for isoniazid resistance is 100 times greater than rifampicin mutation [37].

Castelo-Branco et al. [38] reported certain isoniazid hydrazide compounds labeled 14, 15, and 16 with striking antimycobacterial potency and decreased hepatotoxicity (lower toxicity to HepG2 cells) as compared to the parent isoniazid compounds. Results showed that mycobacterium strains with both isoniazid KatG (SR 0215, SR 2571, and T113) and rifampicin rpoB (T09) mutations were particularly vulnerable to these isoniazid hydrazides. The MIC for 2-Cyano-*N*-(4-methylphenyl) acetamide, 2-Chloro-*N*-(2,6-diethylphenyl) acetamide, and 2-Phenylacetamide were 0.45, 0.43, and 0.47 μM, respectively. There mycobacterial activity is reliant on the isonicotyl attachment and may be contrary to isoniazid molecule. In contrast, isoniazid was mutagenic at 50 μM, while its analogs reflected mutagenicity at concentrations >500 μM, under the same assay conditions. Table 2, represents selected chemotherapeutic derivatives and antimicrobial peptides, their molecular targets, mechanism of action, and investigated MIC range for the promising lead compounds.

Loots [39] reported that isoniazid was less vulnerable to isoniazid-resistant strains by upregulating the fatty acids and alkanes use and synthesizing bioactive moieties specifically pertain to oxidative stress relief, including those linked to a novel degradation mechanism of ascorbic acid. 

Isoniazid, due to its emerging sensitive resistant issues, is going to left out from therapeutic regimens, which endorse the researcher to develop more active novel isoniazid lead derivatives, specifically tailored to target multidrug-resistant tuberculosis. Novel research approaches targeting KatG mutation metabolic schemes and biomarkers have pledged opportunities, which will oblige the discovery of novel isoniazid lead derivatives.

#### 2.2.2. Coumarin Lead Derivatives

The coumarin ring system (chromen-2-one or benzopyran-2-one), existing in natural products that exhibit promising pharmacological properties, has enthralled researchers for decades to probe natural coumarins and their synthetic analogs as potential drug candidates for a wide range of ailments. Coumarins act as intermediate in the synthesis of chromenes, coumarones, furocoumarins, and 2-acylresorcinols [57]. Coumarin and most of its derivatives possess a wide range of therapeutic effects such as antimicrobial [58,59,60,61,62,63], antioxidant [64,65,66], antidepressant [67,68,69], anti-inflammatory [70,71,72,73,74], antitumor [75,76,77,78,79], antinociceptive [80,81,82], antiasthmatic [83,84,85], anti-Alzheimer [85,86,87,88], antipyretic [89,90], and antihyperlipidemic [91,92,93,94] activities. A substantial effort has been made in past few decades to explore coumarin-based entities as antitubercular drug, which is agile against clinically approved therapeutic targets reflecting terrific therapeutic outcomes. Over the years, studies on the antimicrobial function of coumarin-based compounds have increased. Coumarin functional nucleus substituted with varied moieties at all positions has resulted in potent antitubercular activity, except for position 1 and 2 [95].

Coumarins have several biological properties that include the inhibition of *mycobacterium tuberculosis* [96,97,98] and are dependent on structural stability and the scaffolding potential of the parent coumarin compound [45]. Reddy et al. [41] has recently investigated the synthesis of two unique coumarin derivatives, dimethyl substituted compound (1e) and a coumarin-oxime ether derivative (1h), both of which showed excellent antituberculosis activity having MIC of 0.32 and 0.12 μM, respectively. This range is quite comparable to first-line antitubercular drug isoniazid. According to Mangasuli et al. [42], a coumarin-theophylline hybrid (3a) demonstrated strong binding associations with MTB 4DQU enzymes and potential antimicrobial properties over a broad-spectrum inhibiting *Staphylococcus aureus*, *Escherichia coli*, *Candida albicans*, and *Salmonella typhi*. 

Possible antituberculosis mechanism of action manifested by coumarin derivatives implies cytochrome synthesis disruption, suppression of cell proliferation, macrophage activation, and kinase inhibitor [43]. The insertion of other substituents of interest like heterocyclic moieties and oximes, in the coumarin basic nucleus, could anticipate direction for synthesizing potential novel antitubercular lead compounds.

Natural coumarin 6-((3,3-dimethyloxiran-2-yl)-5,7-dihydroxy-8-(2-methylbutanoyl)-4-phenyl-2H-chromen-2-one (1g) exhibited the best anti-*M. tuberculosis* activity with IC_50_ value of 47.4 μM [41]. Among the synthetic coumarin derivatives, compounds 4-Hydroxy-7-methoxy-2H-chromen-2-one (LSPN270), 7-Ethoxy-4-hydroxy-2H-chromen-2-one (LSPN271), 4-Ethyl-2-hydroxy-4,4a-dihydropyrano[3,2-c]-chromen-5(10bH)-one (LSPN476), and 8-Ethoxy-4-ethyl-2-hydroxy-3,4-dihydropyrano[3,2-c]-chromen-5(2H)-one (LSPN484), which showed MIC from 63.4 μM (≤62.5 μg/mL) against the reference strain and selectivity index between 1.07 and 4.27, demonstrated significant activity against MDR *M. tuberculosis* clinical isolates (Figure 1) [44].

According to Pires et al. [44], compounds 1g, LSPN270, LSPN271, LSPN476, and LSPN484 may be considered potential candidates for further studies in developing new antitubercular drugs.

#### 2.2.3. Griselimycin Lead Derivatives

Griselimycin and its variant methylgriselimycin are natural cyclic compounds synthesized from Streptomyces strain DSM 40835 in the 1960s, which exhibit promising antibacterial and antimycobacterial activity against the resistant strains of *M. tuberculosis* [99]. Early studies indicated unfavorable pharmacokinetics of griselimycin, given an edge to rifampicin, which was approved at parallel time frame, while the development of griselimycin as antimycobacterial drug was break off.

It was determined that the cocrystal structures of griselimycin derivatives attach to DnaN, representing that DnaN peptide-binding pockets have been employed by the cyclic part of the griselimycin antibiotics. The appraisal of the self-resistance to griselimycin in *Streptomyces* and griselimycin resistance in mycobacterium exhibited amplification of DnaN gene. The DNA polymerase sliding clamp that tie-up the DNA polymerase to DNA and thus confers replicative enzyme responsible for the DNA repair and replication acceleration in prokaryotes [47].

Rolf Müller and his team determined to upgrade griselimycin, considering its promising bioactivity [48]. Primarily, they enhanced the metabolic stability of griselimycin by the addition of alkylation to the proline residue on position 8. Cyclohexyl griselimycin, a synthetic analog of griselimycin, was metabolically stable and showed enhanced lipophilicity and penetration to the thick cell wall of mycobacterium, resulting in superb antimycobacterial activity, with a MIC value of 0.05 μM against the compound susceptible to the mycobacterium tuberculosis H37Rv strain and 0.17 μM for macrophage-like cell encapsulated same strain (RAW264.7) [47,49]. By attacking the DnaN (sliding clamp that anchors the DNA to DNA polymerase), cyclohexyl griselimycins works against mycobacterium tuberculosis and ultimately prevents the proliferation and regeneration of mycobacterial DNA [47,48]. Besides this, significant antimycobacterial activity in acute and chronic tuberculosis mouse models has revealed its synergistic properties with rifampicin and pyrazinamide, with likelihood of indication for shortening the length of the tuberculosis course [100].

### 2.3. Lead Antimicrobial Peptides

Antimicrobial peptides (AMPs) also known as host defense peptides are distinguished part of the innate immune response acquired among all type of life. They are the positively charged peptides, encompasses about 12-to-50 amino acids residue [101,102,103]. The existence of the positively charged amino acids moieties, like arginine and lysine, provides a net positive charge to AMPs. The AMPs have hydrophilic and hydrophobic regions, responsible for their amphipathic character, enabling their junction with biological cell membranes while retaining aqueous solubility [32,33]. Following their secondary molecular structure, AMPs are categorized into four categories: α-helix, β-sheet, β-hairpin, and linear non-α-helical loop [104,105]. AMPs principally applied their actions by direct interaction with microbial cell membrane. Therefore, antibacterial specificity and selectivity of AMPs over the host are quite remarkable. The physiochemical and structural characteristics, including general charge, charge angle, length, conformation, amphipathicity, hydrophobicity, and solubility, exert considerable effect on the behavior of AMPs [106,107].

Antimicrobial peptides execute an essential role in the generation of innate immune response, having modulating capability of the host cellular immunity. AMP expedite pathogen clearance effective for bacteria, viruses, fungi, and parasites, claiming their broad-spectrum antimicrobial coverage. Peptides have been reported to induce the breakdown or alteration of bacterial cell walls and the cytoplasmic elements condensation. AMPs hit the cell wall by various modes of action, like pore formation, thinning, altered curvature, modified electrostatics, and localized perturbations [108]. It has been reported that the cell membrane and cell wall are the primary cathelicidin-based peptide targets, through inserting osmotic action, cell wall synthesis destruction, and likely DNA binding [109].

AMPs are the emerging investigational agents that play major role in novel drug development process, and some of them are in Phases-II and -III clinical trials, reflecting their significance and trend in the near future. Interestingly, none of the AMP is in clinical trials for the treatment of tuberculosis. Although many research studies reported significant antimycobacterial activity for AMPs, and even considering the preclinical trials of MU1140 (Oragenics, Inc., Tampa, FL, USA), the essential role of AMPs in tuberculosis therapeutics is clearly indicated.

Several in vitro and in vivo studies revealed significant efficacy of AMPs against drug-resistant strains of mycobacterium. A proline-arginine-rich AMP, PR-39, has proved to be effective against clinical isolates of multidrug-resistant strain of *mycobacterium tuberculosis* [110,111]. Fattorini et al. [112] confirmed the inhibitory effect of β-defensin-1 and protegrin-1 against drug-resistant *mycobacterium tuberculosis* growth. One other research study investigated the MICs of synthetic peptides from cecropin A and melittin B hybrid against multidrug-resistant strain of tuberculosis [54]. Most of these tested peptides significantly inhibit the growth of multidrug-resistant tuberculosis strain, having MICs comparable to the ones exhibited from susceptible H37Rv strain. AMPs consisting of D-LAK group, having D-amino acids, was also revealed to inhibit multidrug resistance and extended drug resistance strains of tuberculosis, both in vitro and ex vivo [113]. The LL37 (cathelicidin) manifested to have promising efficacy against H37Rv and drug-resistant strains of mycobacterium in the infected mice [53]. Moreover, 1-month treatment regimen of LL37 (32 μg, three times/week/mouse) indicated for the reduction in up to 53% of mycobacterial number. It has been denoted that the expression of LL37 is impelled in macrophages due to mycobacterial infection (Figure 2) [114,115].

A 26 amphipathic peptide residue, D-V13 K, comprising all D-amino acids, has positively charged lysine and valine at position no. 16, as “specificity determinant.” It was identified the most effective analog against *mycobacterium tuberculosis* having MIC value of 15.6 and 11.2 μM against multidrug-resistant and H37Rv strains, respectively. Peptide D1, having positively charged residue of lysine at the center, demonstrated D5 comparable mycobacterial activity against the multidrug-resistant strains of tuberculosis (i.e., 57 μg/mL) and had therapeutic index at 7.4-fold higher than D5 [54]. In comparison, D-LAK120-HP13 and D-LAK120-A demonstrated the optimum potency and did not detect mycobacterium in wells at concentration of 50 μM and above. AMPs’ nonpolar portion has been incorporated into the membrane bilayer and peptide aggregation, resulting in increased permeability, diminish barrier function, cytoplasmic components disintegration, and ultimately cell death [54].

Bacteriocins (Bcns1–Bcn5) group of peptides from different Gram +ev bacteria has been shown to have improved antimycobacterial action against *mycobacterium tuberculosis* [55]. Sosunov et al. [55], reported that Bcn4 exhibit excellent bacteriostatic capacity at concentration of 0.1 μg/mL, which is almost 10-fold lower than rifampicin MIC_50_ value. The mechanism of action taken by Bcn1–Bcn5 against *mycobacterium tuberculosis* is found on the formation of pores in cell membranes, progressing cell death [116,117,118].

The activities of nisin-A and recent biosynthesized hinge derivatives, i.e., nisin-T, nisin-V, and nisin-S, are evaluated for antimycobacterial activity. Both variants indicated an extra antimicrobial activity compared to nisin-A. The nisin-S was reported as the potent one, effective against *mycobacterium*
*tuberculosis* and *mycobacterium avium*, eradicating their growth by a rate of 29% relative to nisin-A [56].

Cathelicidins is a large class of mammalian AMPs that are considered an essential part of the innate immunity, producing rapid and nonspecific response to pathogens. By regulating cytokine proinflammatory responses, calcium influx, and apoptosis, cathelicidins are substantially recommended as an essential, multifunctional immune regulators with ultimate antimycobacterial activity by the virtue of host adaptive immune responses. The preventive function of cathelicidins in tuberculosis infection depends on the cAMP bursts in macrophages that are activated by mycobacterium, provides tuberculosis pathogenesis clue, and encourages further research into cathelicidin-mediated immune control [52]. NOX_2_ induces phagocytic activity against intracellular bacilli via the formation of reactive oxygen species (ROS). Yuk et al. [119] reported that NOX_2_ is necessary for D3-mediated antimycobacterial action via cathelicidin expression.

To the best our knowledge, studies scrutinizing the antimycobacterial use of AMPs against drug-resistant *mycobacterium tuberculosis* have not still got into preclinical trials. However, considering the emerging trend of AMPs and its essential role in the treatment of drug-resistant tuberculosis, it is rational to anticipate such studies in the very near future.

## 3. Conclusions

Incomplete therapeutic regimen, meager dosing, and the capability of the latent and/or active state tubercular bacilli to abide and to survive against contemporary first-line and second-line antitubercular drugs escalate the prevalence of drug-resistant tuberculosis. It is still a challenge for the pharmaceutical scientists to develop the most promising drug substitute among available favorite drug candidates that potentially cover the resistant strain of *mycobacterium tuberculosis*. In this article, we are fascinated to find the potential drug candidates and their precise tailored derivatives, having promising antimycobacterial properties, but are still on a slow track in clinical research prospective. Our findings suggested the emerging role and scope of antimicrobial peptides and the lead derivatives of coumarin, isoniazid, and griselimycin having strong antimycobacterial activity in laboratories, which need to accelerate their further in vivo studies. Clinical research on these promising compounds possibly leads to potential antitubercular drug candidate that might be more precise and effective against drug-resistant tuberculosis.

## Figures and Tables

**Figure 1 molecules-25-05685-f001:**
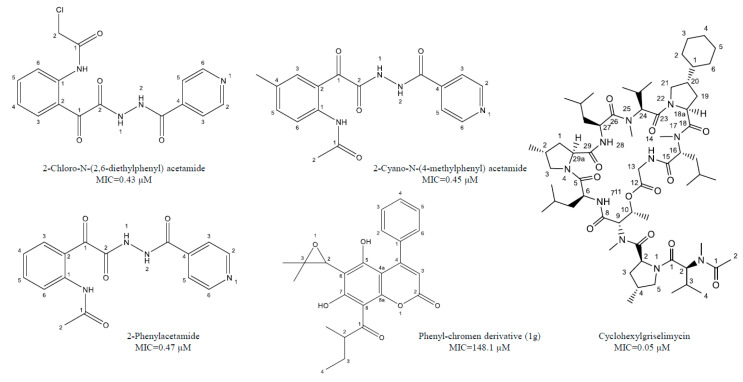
Chemical structure of lead isoniazid, coumarin and griselimycin derivatives with minimum inhibitory concentrations (MICs) value.

**Figure 2 molecules-25-05685-f002:**
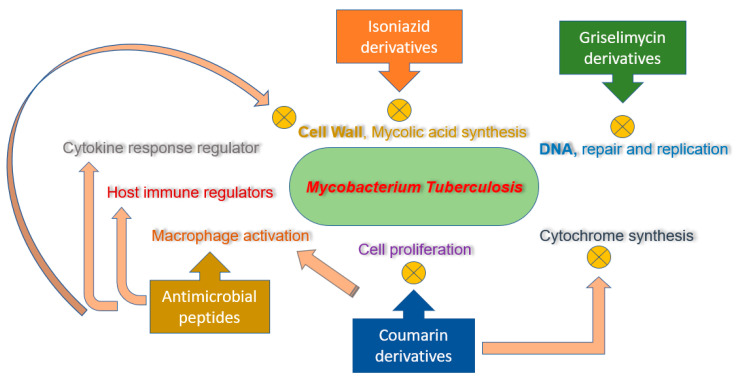
Mechanism of action for isoniazid, coumarin, and griselimycin derivatives and antimicrobial peptides (AMPs).

**Table 1 molecules-25-05685-t001:** WHO classification of antitubercular drugs.

Drug Class	Included Drugs
(A) Fluoroquinolones	Levofloxacin, gatifloxacin, moxifloxacin
(B) Second-line injectables	Streptomycin, kanamycin, amikacin, capreomycin
(C) Other core second-line drugs	Ethionamide, cycloserine/terizidone, prothionamide, linezolid, clofazimine
(D) Noncore, multidrug-resistant tubercular drugs	i. High dose—isoniazid, pyrazinamide, ethambutol ii. Delamanid and bedaquiline iii. Para-aminosalicylic acid, meropenem, cilastatin-imipenem, clarithromycin, clavulanate-amoxicillin

**Table 2 molecules-25-05685-t002:** Selected chemotherapeutic derivatives and antimicrobial peptides, their molecular targets, mechanism of action, and investigated minimum inhibitory concentration (MIC) range.

Drug Class	Lead Compounds	Molecular Targets/Mechanism of Action	MIC Range	References
Isoniazid derivatives	2-Cyano-*N*-(4-methylphenyl) acetamide, 2-Chloro-*N*-(2,6-diethylphenyl) acetamide, 2-Phenylacetamide	Inhibition of mycolic acid synthesis and cell growth inhibitor	0.43–0.47 μM	[38,39,40]
Coumarin derivatives	6-((3,3-dimethyloxiran-2-yl)-5,7-dihydroxy-8-(2-methylbutanoyl)-4-phenyl-2H-chromen-2-one (1g), Dimethyl substituted compound (1e), coumarin-oxime ether (1h), a coumarin-theophylline hybrid (3a), LSPN270, LSPN271, LSPN476, and LSPN484	Cell proliferation inhibitors, cytochrome synthesis disruption, and macrophages activation	0.12–148 μM	[41,42,43,44,45]
Griselimycin derivatives	Cyclohexyl griselimycin	Inhibits DNA repair and replication	0.05–0.17 μM	[46,47,48,49]
Antimicrobial peptides	Bacteriocins (Bcn1–Bcn5), protegrin-1, nisin S, D-V13 K, cathelicidin LL37, D-LAK120-A, and D-LAK120-HP13	Multifunctional host immune regulators, pro-inflammatory cytokine responses regulator, calcium influx, and apoptosis	0.01–30 μM	[50,51,52,53,54,55,56]

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
