# Peer review of "Promising Lead Compounds in the Development of Potential Clinical Drug Candidate for Drug-Resistant Tuberculosis"

_molecules, 2020, doi:10.3390/molecules25235685_

Round 1

Reviewer 1 Report

In this manuscript, the authors aimed at reviewing the lead compounds presently known that are in development as potential drug candidates for the treatment of drug-resistant tuberculosis.

The theme is worthy and requires a constant update. However, to be acceptable, major corrections and improvements should be made.

The introduction is constantly redundant and repetitive and should be improved for easy reading.

Concentrations of substance should be normalized or in micrograms per milliliter or microMolar or both.

Tables should be included that list substances, molecular targets, mechanism of actions, therapeutic regime, references, etc;

Structural formula or organics compounds (and peptide, if possible) should be included;

A schematic draw of molecular targets for the potential drugs mentioned in the text would be meaningful.

The antimicrobial peptide section should be repositioned in the text after or before the chemotherapeutic compounds.

English should be revised for coherence, consistency, spelling, grammar, sentence structures, etc;

Author Response

Reviewer #1

In this manuscript, the authors aimed at reviewing the lead compounds presently known that are in development as potential drug candidates for the treatment of drug-resistant tuberculosis.

The theme is worthy and requires a constant update. However, to be acceptable, major corrections and improvements should be made.

The introduction is constantly redundant and repetitive and should be improved for easy reading.

→ Introduction part is revised, and the repetitive text is removed or linked to the paras, where feel necessary for easy reading and understanding. (P1 L39-41; P2 L48-50; P3 L102-106, L112-115 etc.)

Concentrations of substance should be normalized or in micrograms per milliliter or microMolar or both.

→ Concentrations of substance are normalized to µM, as suggested by the reviewer. (P5 L170-172; P6 L205, 216, 220; P7 L241, 242; P9 L295)

Tables should be included that list substances, molecular targets, mechanism of actions, therapeutic regime, references, etc;

→ The desired table, as Table-2, is added in the revised manuscript. (P5 L185)

 Structural formula or organics compounds (and peptide, if possible) should be included;

→ Structural formula for the lead compounds from isoniazid, coumarin, griselimycin derivatives have been added as “Figure 1” to the main manuscript. (P7 and P8, After L248)

A schematic draw of molecular targets for the potential drugs mentioned in the text would be meaningful.

→ Schematic diagram has been added as “Figure 2” illustrating the molecular targets for selected lead compounds. (P9, After L307)

The antimicrobial peptide section should be repositioned in the text after or before the chemotherapeutic compounds.

→ Antimicrobial peptide section is repositioned in the text after the chemotherapeutic compounds i.e. isoniazid, coumarin, and griselimycin derivatives. (P8, L252 onward)

Thank you for your precious comments.

Reviewer 2 Report

This short review focuses on a limited number of anti-tubercular agents and their development. As such it is of interest as  research  on TB pharmacological treatment certainly deserves high attention. However, what is missing is a clear explanation on why the Authors have selected these specific drugs and why they decided not to cover others that are under development (at different stages) along the drug discovery pipeline.

As it is, therefore, the manuscript cannot be published due to unclear reasons why only addressing these specific medicines. In this respect for instance the title anticipate content of the paper that than is not there. A reader expects to see a wider coverage.

Round 2

Reviewer 1 Report

The present version of the manuscript was improved and acceptable for publication.

Author Response

>>>The present version of the manuscript was improved and acceptable for publication.

>>> The present version of the manuscript was improved and acceptable for publication.

Our Response: Thank you for your kind words and encouragement.

Regards

Reviewer 2 Report

The Author's explanation for the choice of targets to be covered in the review is satisfactory. Accordingly, the title should be changed and adapted. As an example, the word "significant" sounds inappropriate.

Author Response

>>> The Author's explanation for the choice of targets to be covered in the review is satisfactory. Accordingly, the title should be changed and adapted. As an example, the word "significant" sounds inappropriate.

Our Response: Thank you very much for recognizing our efforts and your kind response. As per your precious suggestion, we revised the title by removing the highlighted word "Significant" from the title.

"Promising lead compounds in the development of potential clinical drug candidate for drug-resistant tuberculosis".

Thank you once again for the corrections you have made with us in the final version of our manuscript.

Regards